# The Rational Design and Biological Mechanisms of Nanoradiosensitizers

**DOI:** 10.3390/nano10030504

**Published:** 2020-03-11

**Authors:** Hainan Sun, Xiaoling Wang, Shumei Zhai

**Affiliations:** 1Key Laboratory of Colloid and Interface Chemistry of the Ministry of Education, School of Chemistry and Chemical Engineering, Shandong University, Jinan 250100, Shandong, China; sunhainan1986@126.com (H.S.); wangxiaoling@mail.sdu.edu.cn (X.W.); 2Shandong Vocational College of Light Industry, Zibo 255300, Shandong, China

**Keywords:** nanoradiosensitizers, radiosensitization, radiotherapy, biological mechanisms, physicochemical properties

## Abstract

Radiotherapy (RT) has been widely used for cancer treatment. However, the intrinsic drawbacks of RT, such as radiotoxicity in normal tissues and tumor radioresistance, promoted the development of radiosensitizers. To date, various kinds of nanoparticles have been found to act as radiosensitizers in cancer radiotherapy. This review focuses on the current state of nanoradiosensitizers, especially the related biological mechanisms, and the key design strategies for generating nanoradiosensitizers. The regulation of oxidative stress, DNA damage, the cell cycle, autophagy and apoptosis by nanoradiosensitizers in vitro and in vivo is highlighted, which may guide the rational design of therapeutics for tumor radiosensitization.

## 1. Introduction

Cancer is one of the leading threats to human health [1]. Despite the great advances in cancer biology and clinical treatment in recent years, the survival rate of cancer patients has only slightly improved in recent decades [2]. Various methods, such as surgery, chemotherapy, and radiotherapy, are used clinically in cancer treatment [3,4]. Among these methods, radiotherapy is based on high-energy ionizing radiation to destroy cancer cells and can be used alone or combined with other therapies [5,6]. The intrinsic drawbacks of radiotherapy, such as toxic side effects to the human body and radiation resistance in cancer cells [7,8,9], promote the investigation of radiosensitizers, which can significantly enhance the treatment effect of radiotherapy. 

Due to their outstanding physicochemical properties, nanoparticles (NPs) are widely used in various fields, including the consumer product, energy, and biomedical fields [10,11,12,13]. Various kinds of NPs have been found to sensitize cancer cells to radiotherapy due to their intrinsic radiosensitive action and loading capacity for drugs and siRNAs [14]. Moreover, in vitro and in vivo experiments have demonstrated that NPs exhibit radiosensitization effects by regulating multiple biological mechanisms, such as oxidative stress, DNA damage, cell cycle arrest, apoptosis, autophagy, and hypoxia-related mechanisms [15,16,17].

In this review, we summarize the recent progress made in NPs-induced radiosensitization by modulating multiple biological mechanisms. Hypoxia-related radiosensitive effects have been systematically reviewed by Li and Zhang et al. [18,19] and, therefore, are not addressed in this review. The regulation of crucial cellular processes by the physicochemical properties of NPs in cancer cells is reviewed, which may guide the design of nanoradiosensitizers in the future.

## 2. Oxidative Stress

Reactive oxygen species (ROS) are generated from oxygen in physiological environments mainly in mitochondria and nicotinamide adenine dinucleotide phosphate (NADPH) oxidase. In the mitochondrial respiratory chain, leaked electrons are captured by oxygen, resulting in ROS generation [20]. The activation of NADPH oxidase in the cell membrane also leads to the catalyzed transformation of oxygen into ROS [21]. In addition to these two intrinsic pathways, high-energy ionizing radiation can also elicit ROS generation by directly reacting with water and/or oxygen molecules [22]. The accumulation of ROS in cells leads to oxidative stress [23]. In this section, we summarize the radiosensitization effect of NPs based on this oxidative stress mechanism (Table 1) and the regulation of oxidative stress in cancer cells by the physicochemical properties of NPs.

### 2.1. Nanoradiosensitizers Based on Oxidative Stress

Gold-based nanomaterials have been extensively investigated as a new type of NP-based radiosensitizer involved in cancer radiotherapy. Gold nanoparticles (GNPs) with various decorations can sensitize cancer cells to radiation through the induction of oxidative stress. For example, both polyethylene glycol (PEG)-functionalized GNPs (20 nm) and amino-PEG-functionalized GNPs (10 nm) radiosensitized cancer cells through the induction of oxidative stress [24,25]. Under X-ray irradiation, gold-levonorgestrel nanoclusters (2 nm) consisting of Au_8_(C_21_H_27_O_2_)_8_ (Au_8_NC) with bright luminescence (58.7% quantum yield) and satisfactory biocompatibility promoted the production of ROS and induced cytotoxicity in human esophageal squamous cancer cells (EC1 )cells and EC1 tumor-bearing nude mice [26]. The overexpression of glutathione (GSH) in tumors impaired these radiotherapy effects. Histidine-capped gold nanoclusters (Au NCs@His) (3 nm) improved the radiotherapy effects by depleting intracellular GSH in U14 cells [27].

Enhancement of cellular uptake is beneficial to the radiosensitization effects induced by GNPs. For example, thioglucose decoration on GNPs (14 nm) increased cellular uptake by approximately 31% in human ovarian cancer cells (SK-OV-3) compared with the uptake of naked GNPs, with the former inducing elevated levels of ROS production and enhanced radiotherapy [28]. The modification of GNP-PEG with positively charged cell-penetrating peptides significantly increased GNP internalization and ROS production in the LS180 colorectal cancer cell line, leading to enhanced radiosensitivity [29]. 

Mitochondria-targeting NPs exhibited excellent radiosensitization effects. Peptide (CCYKFR)-templated Au nanoclusters (AuNCs, 3 nm) efficiently targeted mitochondria in human breast cancer MCF-7 cells. CCYKFR–AuNCs irradiated by 4 Gy of X-rays elicited a burst of mitochondrial ROS, resulting in cancer cell death (Figure 1) [30]. In another study, a mitochondria-targeted nanoradiosensitizer was constructed by covalently attaching mitochondria targeting triphenylphosphine (TPP) groups to TiO_2_-Au NPs. When irradiated with X-rays, TiO_2_-Au-TPP NPs (18 nm) triggered ROS production in mitochondria, caused a decrease in mitochondrial membrane potential, and induced the death of MCF-7 cells. In a MCF-7 and 4T1 tumor-bearing Balb/c mouse model, TiO_2_-Au-TPP dramatically improved the therapeutic efficacy of radiotherapy [31].

The surface physicochemical properties of GNPs can modulate this radiosensitization effect. Gold nanospikes (GNSs, 54 nm) were decorated with different ligands, including HS-PEG-CH_3_ (GNSs), HS-PEG-NH_2_ (NH_2_-GNSs), HS-PEG-folate (FA-GNSs), and cell-penetrating peptide TAT (TAT-GNSs). These GNSs exhibited the following radiosensitization effects in human oral squamous KB cells in descending order of magnitude: TAT-GNSs > FA-GNSs > NH_2_-GNSs > GNSs; these effects are correlated dosely with their levels of cellular uptake. Among these GNSs, the sensitization enhancement ratio (SER) for the TAT-GNSs reached 2.30. The radiosensitization mechanisms of these GNSs involved increased ROS generation and mitochondrial depolarization. Radiotherapy of U14 tumor-bearing mice in vivo demonstrated that TAT-GNSs exhibited the best radiosensitization effect [32]. Zwitterionic glutathione-coated AuNPs (GS-AuNPs) sensitized MCF-7 cells to radiation delivered with a clinically appropriate megavoltage photon beam at a low dose of approximately 2.25 Gy, while PEG decoration exhibited protective effects against radiation. The ligand effect on radiosensitization is thought to be independent of cellular uptake but may be related to ROS regulation [33]. 

The shape of gold (Au) nanomaterials can also regulate the efficiency of radiosensitization. GNPs (53 nm), GNSs (54 nm), and gold nanorods (GNRs, 50 nm × 12 nm) with the same PEG coatings exhibiting different radiosensitization effects in KB cells, the effects of which followed the same order as cellular uptake level (GNPs > GNSs > GNRs). The radiosensitization mechanisms involve ROS production, with GNPs inducing the highest ROS and cytotoxicity levels under irradiation [34]. 

The hybridization of GNPs with other NPs also exhibited excellent radiosensitization effects. Fe_3_O_4_-Au nanoparticles (Fe_3_O_4_-Au pNPs, 12.5 nm) significantly enhanced ROS levels in MCF-7 and A549 cells and sensitized both kinds of cells to X-rays. However, in MCF-10A noncancer cells, Fe_3_O_4_-Au pNPs only slightly promoted the production of ROS and ultimately failed to exhibit a radiosensitization effect [35]. *α*-Fe_2_O_3_ coated with ultrasmall gold nanoseeds (*α*-Fe_2_O_3_@Au, 49 nm) sensitized 4T1 cells to radiotherapy through ROS generation. Near-infrared (NIR) treatment induced ablation of *α*-Fe_2_O_3_ and the aggregation of GNPs of approximately 13 nm, which was the optimal size for ROS production and radiotherapy [36]. GNRs (61 nm × 17 nm) encapsulated with Se shells (Au@Se-R/A NCs, R/A: arginine-glycine-aspartic acid (RGD)/activatable cell-penetrating peptide (ACPP), 120 nm) were used as sensitizers in radiochemotherapy by synergizing the radiosentization effects of GNRs and the anticancer effects of the Se shells. The radiosentization mechanism of the Au@Se-R/A NCs involved ROS overproduction [37]. Dumbbell-like Au-TiO_2_ NPs (DATs, 70.1 ± 4.9 nm) showed a synergistic therapeutic effect on radiation therapy, mainly because of strong asymmetric electric coupling of the metals with high atomic number, and the dielectric oxides at their interfaces. DATs significantly sensitized SUM159 triple-negative breast cancer (TNBC) cells to X-rays through oxidative stress and reduced tumor volume in SUM159-tumor-bearing mice, increasing their median survival [27].

In addition to gold-based NPs, other kinds of NPs can enhance radiation sensitivity. AgNPs induced radiation sensitivity by decreasing the levels of catalase (CAT), superoxide dismutase (SOD), and total GSH in human hepatocellular carcinoma HepG2 cells [38]. The induction of ROS played an essential role in the radiosensitization of AgNPs (15 nm) in human U251 glioma cells [39]. TNBC cells are more vulnerable to reagents that elicit oxidative stress than are non-TNBC cells. AgNPs (20–30 nm) were found to induce higher levels of oxidative damage in TNBC cells than they did in non-TNBC cells, leading to the reduction of TNBC growth and improvement of radiation therapy [40].

Iron-based NPs can also sensitize cancer cells to radiation through an oxidative stress mechanism. For example, magnetic iron-oxide NPs (IONPs, 10 nm) decorated with the epidermal growth factor receptor (EGFR) antibody cetuximab were targeted EGFRvIII-overexpressing glioblastoma (GBM) cells. A significant antitumor effect was found in vitro after treatment with cetuximab-IONPs and ionizing radiation. Moreover, in vivo experiments indicated that the overall survival of nude mice was significantly increased after the combination treatment described above was administered to them. These radiosensitization mechanisms involve ROS production [41]. The spherical aggregates of PEGylated ferrocene (Fc-PEG, 75 nm) sensitized 4T1 cells to X-rays through the enhanced generation of ROS [42].

SeNPs (27.5 nm) induced both endogenous and radiation-induced ROS formation in MCF-7 cells under irradiation [29]. The surface decoration regulated the radiosensitization effects of SeNPs. Specifically, SeNPs (200 nm) decorated with chitosan and transferrin significantly enhanced the radiotherapy effects of ^125^I seeds through the activation of ROS production and p53-mediated apoptotic pathways in C6 and A375 cells [43]. In another study, PEG-SeNPs sensitized HeLa cells to radiotherapy through an oxidative stress mechanism. However, polyvinylpyrrolidone (PVP)-SeNPs did not exhibit radiosensitization effects [44].

A series of metal oxide NPs exhibited radiosensitization effects based on an oxidative stress mechanism. For example, gadolinium oxide NPs (3 nm) induced hydroxyl radical production and oxidative stress in a dose- and concentration-dependent manner under X-ray irradiation in non-small-cell lung cancer (NSCLC) cells [45]. Gd_2_O_3_ and CeO_2_-Gd NPs (<100 nm) sensitized U-87 MG cells to radiation through induced oxidative stress [46]. Gd-doped titania NPs ((TiO_2_(Gd) NPs), 20 nm) decorated with 4-carboxybutyl triphenylphosphonium bromide (TPP) targeted mitochondria in MCF-7 cells. TiO_2_(Gd) NPs boosted ROS production in mitochondria under X-ray irradiation and greatly amplified the antitumor efficacy of radiotherapy [47]. Cerium oxide NPs increased ROS production in pancreatic cancer cells, resulting in the activation of thioredoxin 1 (TRX1)-apoptosis signaling kinase 1 (ASK1)-c-Jun terminal kinase (JNK) (TRX1-ASK1-JNK) redox-sensing pathway and apoptosis [48]. In contrast, another work showed that cerium nanoparticles (CNPs) exhibited higher SOD-mimetic activity, which could be modulated by the change in the anion of the precursor salt [55]. CuO NPs (5.4 nm) significantly increased oxidative stress levels, resulting in radiosensitization effects in MCF-7 cells and U14 tumor-bearing nude mice [49].

SiNPs (<5 nm) significantly enhanced ROS production in rat glioma C6 cells under X-ray irradiation. In the absence of SiNPs, ROS levels were not enhanced upon X-ray irradiation [50]. Positively charged NH_2_-SiNPs penetrated the mitochondrial membrane and significantly increased intracellular ROS levels in MCF-7 cells under X-ray irradiation; however, uncapped SiNPs did not increase intracellular ROS levels [51]. 

Carbon-based NPs can also induce radiosensitization effects based on an oxidative stress mechanism. For example, the combination of nano-C_60_ (90–100 nm) administration and ^60^Co *γ* irradiation induced higher ROS levels and enhanced cytotoxicity in B16 and human hepatocellular carcinoma SMMU-7721 cells than did ^60^Co treatment alone [52]. Hydrogenated nanodiamonds (16 nm) with a positive charge enhanced the radiotherapy effects in three radioresistant cancer cell lines (Caki-1, ZR75.1S, ZR75.1R). These radiosensitization mechanisms involved oxidative stress [53]. NH_2_-decorated multiwalled carbon nanotubes (NH_2_-MWCNTs) were loaded with ruthenium polypridyl complex (RuPOP@MWCNTs) via *π*-*π* and hydrogen bond interactions. The positive charge on the MWCNTs promoted NP cellular uptake into cancer cells. RuPOP@MWCNTs significantly enhanced the radiation effects of clinically appropriate X-ray irradiation of drug-resistant R-HepG2 cells through an oxidative stress mechanism [54].

### 2.2. The Impact of the Nanoparticles’ (NP) Physicochemical Properties on Oxidative Stress

Generally, the size of NPs is negatively correlated with the oxidative stress level induced in cancer cells. Triphenylphosphine monosulfonate (TPPMS)-GNPs (1.4 nm) induced higher ROS levels than their 15 nm-sized counterparts in HeLa cells [56]. GNPs of different sizes (30, 50, 90 nm) regulated oxidative stress levels in HL-60 and HepG2 cells, with GNPs of 30 nm treatment resulting in the lowest GSH level, followed by that induced by GNPs of 50 nm and 90 nm [57]. GNPs of 5 nm elicited the highest ROS level in HepG2 and L02 cells, followed by 20 nm- and 50 nm-sized GNPs [58]. The intracellular ROS level induced by PEG-GNPs (6.2–61.2 nm) was also negatively correlated with NP size in HepG2 and HeLa cells [59]. In addition to GNPs, NPs of other sizes were also negatively correlated with oxidative stress levels in cancer cells, such as silica NPs [60,61], AgNPs [62], PVP-AgNPs [63] and Cu_2-*x*_Se NPs [64].

The shape of NPs can regulate oxidative stress levels in cancer cells. Rod-like NPs induce higher levels of ROS in cells than do spherical NPs [65,66], while octahedral-shaped Cu_2_O NPs induce higher ROS levels and cytotoxicity than hexagonal or cubic Cu_2_O NPs [67]. The surface charge of NPs can regulate the oxidative stress level in cancer cells. We and other researchers have found that the intracellular ROS level is positively correlated with positively charged NPs, such as GNPs [68,69,70], polyethylenimine (PEI)-decorated GNRs [71], single-walled carbon nanotubes (SWCNTs) [72], ZnO NPs (98 nm, 11.1 mV) [73], upconversion (UCNP)@SiO_2_ NPs with NH_2_ decoration (37 nm, 26.2 mV) [74], and glucose-decorated iron oxide NPs (70 nm) [75]. In addition, we found that, in A549 cells, hydrophobic GNPs are more likely to induce oxidative stress than are hydrophilic NPs [68]. Furthermore, the length of the hydrophobic moieties in positively charged ligands was found to be positively correlated with intracellular ROS levels induced in HeLa cells treated with GNPs (2 nm) [76]. 

## 3. DNA Damage

DNA in cells continually incurs various types of damage, and cells have devised ingenious mechanisms to repair DNA damage [77]. DNA damage leads to many kinds of diseases, including cancer [78]. However, DNA damage induced by radiation also plays key roles in cancer treatment [79]. In this section, we summarize the radiosensitization effect of NPs on DNA damage mechanisms (Table 2). The regulation of DNA damage in cancer cells by physicochemical properties of NPs is also summarized because it is beneficial to the design of nanoradiosensitizers.

### 3.1. Nanoradiosensitizers Based on DNA Damage

The interaction between GNPs and DNA regulates radiosensitization effects. GNPs (<5 nm) can bind to plasmid DNA through electrostatic interactions, resulting in DNA single-strand breaks (SSBs) and double-strand breaks (DSBs) under irradiation [105]. The distance between GNPs and DNA can affect the radiosensitization effect because GNPs with the shortest possible linker induce the greatest radiosensitization effect [106].

GNP-induced DNA damage contributes to the radiosensitization effect on cells. GNPs (2 nm) sensitized MDA-MB-231 breast cancer cells to radiation through the induction of DNA damage [80]. PEG-decorated GNPs (12 nm) increased cellular DNA damage under irradiation in human GBM-derived cell lines and enhanced the survival of orthotopic GBM tumor-bearing mice [81]. Gold and superparamagnetic iron oxide nanoparticles (SPION)-loaded micelles (GSMs, 100 nm) in combination with radiotherapy led to an ~2-fold increase in DSBs in GBM cells [82]. The combination of Au@Se NPs (120 nm) treatment with irradiation induced DNA damage by enhancing ROS generation and the phosphorylation levels of related proteins, including Ataxia-telangiectasia mutated (ATM), Checkpoint kinase 2 (Chk2), Breast cancer susceptibility protein 1 (BRCA1), p53 and histones [37].

The inhibition of DNA repair induced by GNPs also contributes to the radiosensitization effect. For example, homologous recombination (HR) and non-homologous end joining (NHEJ) are intrinsic DSB repair pathways. PEGylated nanogels containing GNPs inhibited the expression of HR and NHEJ-related proteins, including Rad51 and Ku70, resulting in the suppression of the repair of radiation-induced DSBs in murine squamous carcinoma SCCVII cells [83]. DNA-GNPs (4.5 nm) effectively abrogated the repair of radiation-induced DNA DSBs and sensitized U251MG-P1 cancer stem cell-like cells to radiotherapy through the induction of mitotic catastrophe [84]. 

GNPs conjugated with anticancer drugs exhibited a synergistically effective cancer treatment by effectively delivering drugs to tumor cells and enhancing cell radiosensitization activity, such as through DNA damage induction. For example, the combined treatment of cisplatin-tethered GNPs (50 nm) and irradiation significantly enhanced DNA DSBs, as evidenced by the enhanced density of *γ*-H2AX foci, resulting in the apoptosis of patient-derived treatment-resistant glioblastoma multiforme (GBM) cells (Figure 2) [85]. HER-2-targeted Herceptin-GNPs (30 nm) in combination with X-rays induced higher levels of DNA DSBs in HER2-positive human breast cancer cells than X-ray irradiation alone [86,87]. Furthermore, the combined treatment of Herceptin-GNPs and X-ray irradiation significantly inhibited tumor growth compared to X-ray irradiation alone [87]. Doxorubicin (DOX)-loaded GNPs (5 nm) with dual-targeting decoration were more likely to be internalized by HeLa cells than were monotargeting or nontargeting decorations, leading to higher levels of DNA DSBs under irradiation [88]. Chitosan-capped GNPs (18 nm) loaded with doxorubicin (CS-GNPs-DOX) enhanced the chemoradiotherapeutic effect by significantly decreasing cancer cells viability by increasing DNA DSBs in MCF-7 breast cancer cells [89]. Goserelin-decorated pegylated GNRs (gGNRs, 8 nm) were preferentially internalized by PC3 cells through a gonadotropin-releasing hormone receptor-mediated mechanism. The combined treatment of gGNR and irradiation induced higher levels of *γ*-H2AX foci than the combination of pegylated GNR and irradiation [90]. 

The size and surface chemistry of GNPs can regulate the radiosensitization effect through the generation of DNA damage. Citrate-coated GNPs (6, 10, 25 nm) induced different levels of plasmid DNA damage under irradiation. The number of SSB increased as the size of the GNP was decreased [107]. The uptake level of positively charged GNPs (14 nm) was higher than that of negatively charged GNPs by A712 human glioblastoma cells. Upon irradiation, positively charged GNPs induced higher levels of DNA damage and apoptosis than did their negatively charged counterparts [91]. The surface chemistry of the GNPs (32 nm) regulated the DNA damage-mediated radiosensitization effects. Citrate decoration on GNPs induced the highest level of plasmidic DNA damage under irradiation, followed by PEG1000, human serum albumin (HSA), and PEG4000 decoration, for which the level of DNA damage was correlated with hydroxyl radical (HO·) production [108]. Both GSH- and bovine serum albumin (BSA)-decorated Au_25_NCs (<2 nm) with biocompatible coating surfaces preferentially accumulated in tumors via an improved enhanced permeability and retention (EPR) effect, which led to a greater enhancement of cancer radiotherapy than was induced by the much larger Au NPs. Under irradiation, GSH-coated Au_25_NCs induced more significant DNA damage, decreased tumor weight and showed more efficient renal clearance than did BSA-GNPs [92]. 

In addition to gold-based NPs, other metal NPs can induce DNA damage-related radiosensitization effects. For example, AgNPs (20–30 nm) induced more DNA damage with concurrent radiation treatment in triple-negative MDA-MB-231 cells than they did in non-triple-negative MCF-7 and MCF-10A cells, with the effect based on an oxidative stress mechanism and resulting in the inhibition of tumor growth [40]. In another study, AgNPs (20 nm) inhibited the expression of DNA damage/repair proteins, including Rad51, Ku-70, and Ku-80, in nasopharyngeal carcinoma epithelial (CNE) cells under irradiation. The targeted epidermal growth factor receptor-specific antibody decoration on AgNPs can significantly enhance radiation-induced DNA damage, possibly by elevated cellular uptake [93]. Arg-Gly-Asp (RGD) andtransactivator of transcription (TAT) decoration on iridium (Ir) NPs (<5 nm) accumulated in cancer cells and showed cell-nucleus targeting. RGD-Ir-TAT NPs elicited ROS overproduction and DNA lesions in 4T1 cells under X-ray irradiation and exhibited satisfactory destruction of tumor tissues [94].

Metal oxide NPs exhibited radiosensitization effects through DNA damage or the inhibition of DNA repair. Thulium oxide NPs (40–45 nm) enhanced DSBs in radioresistant 9L brain gliosarcoma cells under irradiation [98]. The combined chemoradiotherapy of DOX-loaded mesoporous tantalum oxide (mTa_2_O_5_, 80 nm) NPs led to a strong synergistic therapeutic effect in a mouse tumor model. The interaction of Ta with X-rays induced significant DNA damage during radiotherapy [99]. ZnO NPs (7 nm) sensitized SKLC-6 lung cancer cells to radiation through the induction of DNA damage; however, ZnO NPs exhibited no radiosensitization effect on MRC-5 normal lung cells [100]. Gadolinium-doped ZnO NPs (9 nm) impaired DNA repair by downregulating the mRNA levels of *XRCC2* and *XRCC4* genes under irradiation and induced apoptosis in SKLC-6 lung carcinoma cells [95]. Oleic acid decorated iron-oxide NPs (MN-OA, 10 nm) downregulated proteins involved in DNA double-strand break repair, such as RAD51 and BRCA1, resulting in DNA damage in mouse fibrosarcoma WEHI-164 cells under irradiation [96]. Apurinic endonuclease 1 (Ape1) is an enzyme involved in base excision repair. The SPION (4–6 nm)-based siRNA delivery system knocked down the expression of Ape1 and sensitized brain tumor cells to radiotherapy [97,109].

A series of polymer NPs can be used as drug carriers to enhance the radiosensitization effect by promoting DNA damage. Polymeric NPs containing camptothecin (CRLX101, 20–30 nm) promoted the formation and persistence of radiation-induced DSBs and inhibited radiation-induced HIF1α activation, which resulted in enhanced radiosensitization of HT-29 cells and xenograft models [101]. Irradiation can induce site-specific expression of receptors in tumor cells, such as tax-interaction protein 1 (TIP-1). TIP-1-targeted polymer NPs (<100 nm) loaded with JNK inhibitor molecules significantly inhibited DNA repair in Lewis lung carcinoma (LLC) cells under irradiation and induced greater apoptosis and inhibition of tumor growth compared to irradiation alone [102]. The application of DNA double-strand repair inhibitors (DSBRIs) is a promising strategy to improve radiotherapy. KU55933, a DSBRI, was loaded into PLGA NPs (87 nm). The resulting NP KU55933 improved the radiosensitization of H460 cells and tumor tissues through the downregulation of ATM and AKT phosphorylation [74]. EGF-decorated PLGA NPs (130–140 nm) incorporating a ruthenium-based radiosensitizer preferentially bound to EGFR-overexpressing oesophageal cancer cells and exhibited radiosensitization effects through the induction of DNA damage [110]. Folate-decorated PEI NPs were used to construct a new class of DNA damage repair inhibitors, nanoparticle Dbait (NP Dbait, 140 nm), which were internalized by prostate cancer cells overexpressing folate receptors. Dbait in the nucleus inhibited DNA damage repair signaling pathways by mimicking DNA DSBs, resulting in the activation of DNA-PK and H2AX phosphorylation. DNA damage repair factors were assembled at the end of Dbait and sequestered away from the real DSB sites, resulting in defects in DSB repair in cells under irradiation [103]. X-ray repair cross-complementing protein 1 (XRCC-1) is overexpressed in X-ray-resistant HeLa cells and is critical for the inhibition of DNA repair. Folate decorated-BSA NP (255 nm) loaded with organic selenocompounds increased ROS overproduction and inhibited XRCC-1 expression in HeLa cells under irradiation [104].

### 3.2. The Impact of NP Physicochemical Properties on DNA Damage

Generally, the size of NPs is negatively correlated with DNA damage level. Small GNPs (5 nm) induced DNA damage in HepG2 cells and clastogenic damage in vivo, while larger GNPs (20 nm, 50 nm) did not induce these effects [111]. AgNPs (4.7 nm) induced higher genotoxicity in HepG2 and HL-60 cells than did AgNPs (42 nm), as evidenced by DNA strand breaks and oxidative DNA damage [112]. Small silica NPs (19 nm) induced higher DNA damage levels in HepG2 cells than did larger NPs (43 nm, 68 nm) [61].

The shape of NPs can regulate DNA damage. MWCNTs (10–30 μm/8–15 nm, 0.5–2 μm/8–15 nm) induced single-strand DNA damage and elevated DNA repair gene levels in HepG2 cells, while MWCNTs (10–30 μm/20–30 nm) caused no damage to DNA [113]. Another study reported that the length and diameter of the MWCNTs were positively correlated with the DNA damage level in A549 cells [114]. Needle-shaped PLGA-PEG NPs (30 × 540 nm) caused DNA fragmentation and cytotoxicity in HepG2 cells, while no DNA damage was found in HepG2 cells treated with spherical NPs (90 nm) [115]. In addition to the size and shape of NPs, the NP surface charge can also regulate DNA damage. GNPs (3.1 nm, 24.5 mV) with a positive charge induced the highest DNA damage level in A549 cells, followed by negatively charged GNPs [116].

## 4. Cell-Cycle Arrest

The cell cycle contains four phases: The G1, S, G2 and M phase. The G2/M phase is the most sensitive phase to radiation. Therefore, NPs that can induce cell-cycle arrest at the G2/M phase enhance the radiotherapeutic effect on cancer cells. In recent years, various kinds of NPs were found to exhibit radiosensitization effects through the induction of cell-cycle arrest. They are summarized in this section (Table 3).

### 4.1. Nanoradiosensitizers Based on Cell-Cycle Arrest

GNPs can sensitize cancer cells to radiation through the induction of cell-cycle arrest. For example, the combined treatment of GNPs (5 nm) and neutron/*γ* irradiation induced cell-cycle arrest in the G2/M phase, resulting in the inhibition of migration and invasion of Huh7 and HepG2 cells [117]. The combined treatment of irradiation and GNPs (50 nm) significantly increased the proportion of melanoma cells in the G2/M phase, which enhanced the next radiation treatment [118]. Glucose-GNPs (Glu-GNPs, 11 nm) induced the G2/M arrest of radiation-resistant DU-145 human prostate cancer cells through activation of checkpoint kinases CDK1 and CDK2, resulting in these cells being sensitized to ionizing radiation [119]. The decoration of GNPs with targeting moieties can enhance the radiosensitization effect. Arg-Gly-Asp peptide (RGD)-GNRs sensitized melanoma A375 cells exposed to radiation through the downregulation of radiation-induced integrin *α*_v_*β*_3_ and induction of cell-cycle arrest in the G2/M phase [120]. EGFR antibody-decorated hollow gold nanospheres (anti-EGFR/HGNs, 55 nm) were more efficiently internalized by HeLa cells than were naked HGNs. More cells were arrested in the G2/M phase after induction by anti-EGFR/HGNs than they were after induction by naked HGNs. The combination of anti-EGFR/HGNs and megavoltage irradiation significantly enhanced the number of apoptotic cells than did irradiation alone [121].

GNP size affected the radiosensitization effects of cell-cycle arrest. For example, the size of Glu-GNPs regulated the radiosensitization effects in MDA-MB-231 cells. Glu-GNPs (49 nm) were internalized more efficiently than their counterparts (16 nm) and induced higher levels of G2/M cell-cycle arrest under irradiation [54]. Cho et al. reported that the surface structure of NPs played key roles in enhancing these radiosensitizing effects. They found that day-flower-like nanoparticle (D-NP), which has a large surface area, induced more ROS production in and G2/M stage arrest of HepG2 cells and exhibited significant radiosensitization effects. However, spherical night-flower-like nanoparticle (N-NP), which has a small surface area, did not affect the cell-cycle distribution and exhibited no radiosensitization effects [122].

In addition to GNPs, other NPs also exhibit radiosensitization effects through the induction of G2/M arrest. Graphene quantum dots (GQDs) sensitize colorectal carcinoma cells to ionizing radiation through the induction of ROS generation, G2/M stage arrest, and apoptosis [123]. Hyaluronic acid-functionalized bismuth oxide nanoparticles (HA-Bi_2_O_3_ NPs, 45 nm) exhibited excellent biocompatibility and radiosensitization effects in SMMC-7721 cells and tumor-bearing mice. The combined treatment of HA-Bi_2_O_3_ NPs and irradiation drove a higher proportion of cells into the G2/M phase than did irradiation alone [124]. HSP90 was found to play key roles in the radiosensitization effects of oleic acid-decorated iron-oxide NPs (MN-OA, 10 nm) on WEHI-164 cells and tumor tissues. The interaction between MN-OA and HSP90 led to the downregulation of proteins involved in cell cycle progression, such as cyclin B1 and CDC2, ultimately resulting in G2/M stage arrest [96]. Gadolinium oxide NPs (2–5 nm) sensitized three NSCLC cell lines (A549, NH1299, and NH1650) to carbon ion radiation through the induction of cell cycle arrest in the G2/M phase and cytotoxicity [125]. Titanate nanotubes (TiONts, 10 nm in diameter) were internalized by human glioblastoma cells (SNB-19 and U87 MG cell lines) through endocytosis and diffusion mechanisms. Upon irradiation, TiONts enhanced G2/M cell cycle arrest [126].

NPs loaded with cell-cycle blockers and anticancer drugs can enhance the radiosensitization effects on cells. 7-Ethyl-10-hydroxy-camptothecin (SN-38) induced G2/M cell-cycle arrest. Mesoporous TiO_2_ NPs (45 nm) carrying SN-38 and nucleus-targeting moieties (MTiO_2_(SN-38)-TAT-RGD) accumulated in the nucleus of 4T1-Luc cells and induced G2/M arrest. The combination of MTiO_2_(SN-38)-TAT-RGD and X-ray irradiation inhibited cell proliferation in vitro and decreased tumor volume in vivo (Figure 3) [127]. Folate-decorated hydroxycamptothecin (HCPT)-loaded micelles (HFOL, 132 nm) were effectively internalized by HeLa cells. In vivo experiments indicated that HFOL induced the G2/M phase arrest of tumor tissue cells and inhibited tumor growth upon irradiation [128]. Ceria NPs loaded with the anticancer drug neogambogic acid (NGA-CNPs, 3–5 nm) induced the G2/M phase cell cycle arrest of MCF-7 cells and significant radiosensitization effects [129]. Paclitaxel (PTX)-loaded micelle NPs (NK105, 85 nm) induced more severe G2/M stage arrest and higher radiosensitization of Lewis lung carcinoma cells than did PTX alone [130]. In another study, PTX-loaded poly(D,L-lactide-co-glycolide) (PLGA, 200–500 nm) NPs sensitized HepG2 and HeLa cells to radiotherapy through the induction of G2/M stage arrest [131]. Coculture of MCF-7 cells with paclitaxel-loaded PLGA NPs (500 nm) demonstrated that released paclitaxel blocked cells in the G2/M phase and sensitized MCF-7 cells to radiation [132]. Docetaxel-loaded PLGA NPs sensitized A549 and CNE-1 cells to radiation through enhanced G2/M arrest and apoptosis [133]. The Fe_3_O_4_@ZnO nanocomposites functionalized with transferrin receptor antibody (TfR Ab) delivered DOX into hepatocellular carcinoma SMMC-7721 cells, resulting in G2/M cell cycle arrest in combination with irradiation. In vivo studies demonstrated that tumor growth was significantly inhibited after radiotherapy mediated by Fe_3_O_4_@ZnO/DOX/TfR Ab [134].

### 4.2. The Impact of NP Size on Cell-Cycle Arrest

Citrate-GNPs of 56, 33, and 15 nm regulated the proportion of HepG2 cells in the G2/M phase, with 15 nm citrate-GNPs inducing the highest level of G2/M cell-cycle arrest [135]. ZnO NPs of 20 nm enhanced the proportion of HeLa cells in the G2/M phase, while ZnO NPs of 40 nm and 80 nm did not induce G2/M arrest [136]. Therefore, it seems that smaller NPs are more likely to induce G2/M arrest. The impact of NP shape and surface chemistry on G2/M arrest has rarely been reported.

## 5. Apoptosis

There are two types of apoptosis: extrinsic and intrinsic apoptosis. Extrinsic apoptosis is triggered by the activation of death receptor superfamily proteins on the cell membrane. Intrinsic apoptosis is triggered through the endoplasmic reticulum- or mitochondria-related mechanisms [137,138]. The radiosensitization effects of the NPs based on the apoptosis mechanism are summarized in this section (Table 4). Moreover, the regulation of cancer cell apoptosis by NP physicochemical properties is also summarized.

### 5.1. Nanoradiosensitizers Based on Apoptosis

GNPs exhibit radiosensitization effects based on apoptosis. First, GNPs alone in combination with irradiation induce higher apoptosis rates of cancer cells [139,140]. GNPs conjugated with targeting moieties also exhibit excellent radiosensitization effects that culminate in apoptosis induction. For example, folate-conjugated Au@Fe_2_O_3_ NPs (44 nm) exhibited radiosensitization effects in KB cells by elevating the apoptosis rate [141]. EGFR antibody-decorated hollow gold nanospheres (anti-EGFR/HGNs, 55 nm) induced the downregulation of Bcl-2 and the upregulation of caspase 3, Bax, and Bad in HeLa cells under irradiation [121]. RGD decoration significantly increased the cellular uptake of GNRs@mSiO_2_ through a receptor-mediated mechanism in integrin α_V_β_3_-overexpressing breast cancer cells, resulting in an increased apoptosis rate and inhibition of tumor growth under irradiation compared to the apoptosis and growth rates induced by non-decorated GNRs@mSiO_2_ [142]. Cyclic RGD (cRGD)-conjugated GNPs (20 nm) induced higher apoptosis rates in small cell lung cancer NCI-H446 cell tumor-bearing mice than did irradiation treatment alone. The long-term exposure of cRGD-GNPs in combination with radiotherapy significantly inhibited the growth of tumor tissues [143].

The size of the GNP can regulate the radiosensitization effect. Without radiation exposure, BSA-GNPs (8, 50, 187 nm) exhibited no induced cytotoxicity in hepatocellular carcinoma. Under irradiation, small BSA-GNPs induced higher levels of caspase-3 and Bax expression and lower levels of Bcl-2 expression in mouse tumor tissues than larger GNP counterparts [144]. The size of the PEG-GNPs can also regulate the radiosensitization effect. PEG-GNPs with diameters of 12 nm and 27 nm had stronger sensitization effects than did PEG-GNPs with diameters of 5 nm and 49 nm, as indicated by the induction of apoptosis and necrosis, resulting in the inhibition of tumor growth [145].

GNPs loaded with multiple drugs, antibodies, and siRNA exhibit radiosensitization effects. 17-allylamino-17-demethoxygeldanamycin (17-AAG) is an inhibitor of HSP90 that induces the apoptosis of cancer cells. The combined treatment of 17-AAG, GNPs and irradiation induced the highest expression of caspase-3 in HCT-116 cells compared with single treatments or combinations of two treatments [146]. Cisplatin-loaded gold nanoparticles (Au@PAH-Pt/DMMA, 78 nm) aggregated rapidly through electrostatic interactions in the acidic tumor microenvironment. Under irradiation, the aggregated GNPs induced higher apoptosis rates and inhibited tumor growth than were induced by irradiation alone [147]. Cetuximab-GNPs (30 nm) increased the apoptosis rates in A431 tumor xenografts when administered in combination with irradiation [148]. The MET proto-oncogene receptor tyrosine kinase (c-Met) is overexpressed in multiple malignancies, such as the cervical cancer cell line CaSki, and is related to the metastasis, proliferation, and invasion of cancer. The combined treatment of anti-c-Met antibody-loaded hollow GNPs (56 nm) induced the overexpression of caspase-3 and Bax, resulting in a higher apoptosis rate than did irradiation alone [149]. Head and neck squamous cell carcinoma exhibits an inherent anti-apoptotic mechanism through the upregulation of the sphingosine kinase (SphK1) gene. GNRs (50–70 nm × 35 nm) loaded with siRNA of the SphK1 gene significantly enhanced the expression of caspase 3 under irradiation in human squamous cell carcinoma xenografts [150]. 

In addition to GNPs, other metal-based NPs can also exhibit radiosensitization effects through apoptosis mechanisms. Citrate-AgNPs (15 nm) exhibited powerful radiosensitizing ability by eliciting high apoptosis rates of U251 cells under radiation [151]. The combination of gadolinium-based NPs (3 nm) with irradiation significantly inhibited tumor growth accompanied by an enhanced number of cells in late apoptosis [152]. PEGylated selenium NPs (80 nm) combined with X-ray irradiation exhibited higher caspase-3 activity and apoptosis in A549 cells than did irradiation or PEGylated selenium NPs alone [153]. The combination of QDs (48 nm) and irradiation induced higher levels of cleaved caspase 3 and apoptosis in H460 cells than did irradiation alone [154]. Cu_2_(OH)PO_4_ NPs (5 nm) generated Cu^I^ sites under irradiation, which catalyzed the decomposition of H_2_O_2_ into hydroxyl radicals in the tumor microenvironment, resulting in the apoptosis and necrosis of HeLa cells [155]. Cationic nanoscale metal–organic frameworks (nMOFs, <100 nm) containing Hf and Ru elements were targeted to mitochondria, resulting in significant depolarization of the mitochondrial membrane, an increased apoptosis rate, and inhibition of tumor growth in mouse models under irradiation (Figure 4). 

Non-metal NPs also exhibit radiosensitization effects. Valproic acid (VPA) sensitized multiple cancer cell types to radiation through the inhibition of histone deacetylase (HDAC). VPA-loaded mesoporous silica NPs (<200 nm) significantly upregulated caspase-3, p53, and cleavage of poly (ADP-ribose) polymerase (PARP) and downregulated Bcl-2, resulting in the apoptosis of C6 and U87 cells under radiation [157]. In another study, the combination of HER2-targeted silica NPs (40 nm) and irradiation induced a higher rate of apoptosis of HER2-overexpressing breast cancer cells than did irradiation alone [158]. Black phosphorus QDs can render tumor cells sensitive to radiotherapy through the overproduction of ROS and elevated apoptosis rates [159]. It was also found that black phosphorus nanosheets (300 nm × 25 nm) enhanced the radiotherapy effects in A375 cells as indicated by higher apoptosis rates and DNA damage levels than were induced by irradiation alone [160]. 

A series of polymer NPs can be used as nanocarriers to deliver genes, drugs or antibodies that sensitize cancer cells to radiotherapy. Dendrimer NPs (20 nm) loaded with recombinant DNA plasmids enhanced the gene expression of tumor necrosis factor *α* (TNF*α*) and herpes simplex virus type 1 thymidine kinase (HSV1-TK) and the apoptosis rates of human choroidal melanomaOCM-1 cells, exhibiting radiosensitization effects under irradiation [161]. The combination of docetaxel-loaded PEG-Pep-PCL NPs (85 nm) and radiation led to higher ROS levels and apoptosis rates in gastric cancer cells than did the combination of docetaxel and radiation or irradiation alone [162]. The MET aberrant activation plays key roles in radiotherapy resistance. The angiogenic inhibitor kringle 1 domain of hepatocyte growth factor (HGFK1) was reported to strongly bind to MET. Cationic copolymer NPs encapsulating a plasmid encoding the *HGFK1* gene promoted the radiation-induced apoptosis of U87 and U251 human glioblastoma cells and inhibited tumor growth in vivo [163]. Heat shock protein 70 (Hsp70) mediates the protection of tumor cells against apoptosis. Hsp70-specific antibody (cmHsp70.1) and survivin-targeting miRNA plasmid-loaded human serum albumin (HSA) NPs (180 nm–220 nm) significantly reduced survivin expression and enhanced caspase 3/7 activity in U87 MG and LN229 glioblastoma cells under irradiation [164].

### 5.2. The Impact of NP Physicochemical Properties on Apoptosis

The size of an NP can determine the apoptosis rates of cancer cells. For example, the apoptosis rate of human colon carcinoma LoVo cells induced by AgNPs (10, 20, 40, 60 and 100 nm) was negatively correlated with NP size [62]. AgNPs (4.7 nm) induced higher expression of caspase 3 and caspase 7 than did AgNPs (42 nm) in HepG2 and HL-60 cells [165]. AgNPs (13 nm) caused a decrease in the anti-apoptotic protein Bcl-2 and an increase in the pro-apoptotic protein Bax and upregulated the phosphorylation of NF-*κ*B in A549 cells, indicating that AgNPs (13 nm) induced apoptosis via the NF-*κ*B pathway. However, large AgNPs (45 and 92 nm) did not activate the apoptosis signaling pathway [166]. The size of silica NPs (19, 43, 68 nm) is negatively correlated with the apoptosis and necrosis rates of HepG2 cells [61]. However, another study showed that silica NPs (50 nm) induced higher apoptosis rates of HepG2 cells than did silica NPs (20 nm) [60]. ZnO NPs (49.4 nm) induced higher apoptosis rates of human neuroblastoma SHSY5Y cells than did ZnO NPs (90.8 nm) [167]. The translocation of Bax from the cytoplasm to mitochondria is a crucial step in apoptosis. Smaller nano-C_60_ were more likely to induce the translocation of Bax than were larger NPs in MCF-7 cells [168]. 

The shape of NPs can also regulate the apoptosis rate of cancer cells. ZnO nanorods (65 × 150 nm) induced higher rates of HeLa and SiHa cell apoptosis than did ZnO nanosphere (60 nm), as evidenced by the reduction of phospho-Bad and PARP cleavage [169]. The aspect ratio of silica NPs can determine the apoptosis rate. Silica NPs (aspect ratio = 4) induced the highest apoptosis rates of A375 cells, followed by silica NPs with aspect ratio = 2 and aspect ratio = 1 [170]. Polystyrene nanospheres (20.6 nm) induced a higher rate of HeLa cell apoptosis than did nanodisks (19.7 nm) [171]. Needle-shaped PLGA-PEG NPs (30 × 540 nm) induced a higher expression of caspase 3 in HepG2 cells than did spherical NPs (90 nm) [115]. In addition, the hydrophobicity of GNPs (20–25 nm) regulated the rate of A549 cell apoptosis. Hydrophobic GNPs induced higher apoptosis rates than did hydrophilic GNPs [172].

## 6. Autophagy

Autophagy is a process related to the degradation of dysfunctional or unnecessary components in cells. Autophagy is categorized into three types: microautophagy, macroautophagy, and chaperone-mediated autophagy [173,174]. Growing evidence has demonstrated that autophagy plays key roles in the pathogenesis of cancer [175]. Moreover, cytostatic autophagy leads to cancer cell death [176]. In this section, we summarize NPs’ radiosensitization effect based on the autophagy mechanism (Table 5). The regulation of autophagy in cancer cells by NPs is also summarized, which may guide the design of nanoradiosensitizers.

### 6.1. Nanoradiosensitizers Based on Autophagy

Multiple kinds of NPs can sensitize cancer cells to radiation based on the autophagy mechanism. The combined treatment of AgNPs and irradiation on hypoxic glioma U251 cells led to autophagy. The inhibition of autophagy by 3-MA alleviated cytotoxicity, indicating that autophagy plays key roles in radiosensitization effects [177]. Fe_3_O_4_@Ag NPs (11 nm) also exhibited radiosensitization effects in U251 cells through the inhibition of protective autophagy and the eventually increase in calcium-dependent apoptosis [178]. Gadolinium oxide NPs (2–5 nm) sensitized NSCLC cells (A549, NH1299, and NH1650) to radiation and induced cytostatic autophagy [125]. Copper cysteamine NPs sensitized SW620 colorectal cells to X-ray irradiation by diminishing the mitochondrial membrane potential and inducing autophagy [179]. CuO NPs (5.4 nm) induced destructive autophagy, revealing a radiosensitization effect in MCF-7 cells and U14 tumor-bearing nude mice [49]. FePt/GO nanosheets inhibited the proliferation of NSCLC H1975 cells and sensitized H1975 cells to radiation. The radiosensitization mechanisms involved ROS production and autophagy [180].

### 6.2. The Impact of NP Physicochemical Properties on Autophagy

The size of NPs can regulate autophagy levels in cancer cells. For example, AgNPs of 10 nm induced higher expression of LC3-II in HepG2 cells, followed by 50 nm and 100 nm AgNPs [181]. The size of palladium nanoparticles (PdNPs, 6 nm, 12 nm, 20 nm) also regulated autophagy in HeLa cells, with 20 nm-sized PdNPs inducing the highest accumulation of autophagosomes through mammalian target of rapamycin (mTOR) signaling pathway inhibition and autophagic flux blockade [17]. Polymeric NPs of 141 nm induced higher autophagy levels in MCF-7 cells than did polymeric NPs (44 nm) [182]. The shape of GNPs can also regulate autophagy levels in cancer cells. Gold nanospheres (20 nm in diameter) induced more autophagosome accumulation in HeLa cells than did GNRs (40 nm in length) through the inhibition of autophagic flux [183].

The surface charge can regulate the autophagy level. The positively charged cetyl trimethyl ammonium bromide (CTAB)-GNRs promoted the transformation of LC3-I to LC3-II in HCT116 cells, while the negatively charged polystyrene sulfonate gold nanorods ((PSS)-GNRs) did not significantly enhance the autophagy level [184]. Similarly, CTAB-GNRs (55 × 14 nm) induced AKT-mTOR-dependent autophagy by inducing LC3-II conversion and p62 degradation in A549 cells. However, PSS-GNRs did not induce autophagy (Figure 5) [185]. NH_2_-GQDs (3.5–5 nm) induced autophagy in A549 cells, as proven by LC3-II conversion and autophagosome accumulation; however, negatively charged COOH-GQDs did not enhance autophagy levels [186]. COOH-SWCNTs induced the autophagic cell death of A549 cells, however, PEG decoration inhibited autophagy [187].

## 7. Conclusions and Perspectives

The combination of nanotechnology with radiation therapy significantly increases the precision of therapy and reduces side effects. NPs of different compositions sensitize cancer cells to radiotherapy through multiple mechanisms, including oxidative stress, DNA damage, cell-cycle arrest, apoptosis and autophagy (Figure 6). For successful radiotherapy, some important strategies are considered. First, NPs with high atomic number (*Z*) are used to enhance radiation therapy efficacy via their photoelectric and Compton effects. Second, targeting cancer cells with specific targeting molecules prolongs the circulation time of the NPs to increase their accumulation in cancer cells. Third, the combination of two different types of radiosensitizers or the combination of radiosensitizers and anticancer drugs or siRNA in a single nanostructure can result in significantly synergistic tumoricidal effects. Finally, the anti-tumor immune response induced during nanotechnology-based radiosensitization has shown great effect in reducing the side-effects of radiation [188] and enhancing the abscopal effects of radiationtherapy [189], which present novel strategies to design radiosensitizers.

Despite rapid advancement over the past years, more attention could be paid to the following issues. The radiosensitization effects of NPs are strongly related to their preparation procedures, particle sizes, geometries, surface chemistries and biosafety. At present, most of the nanoradiosensitizers are developed with high-*Z* metal elements, which are usually undegradable and cause biosafety concerns. Therefore, the impact of the physicochemical properties on the biocompatibility, biodistribution, biodegradability and clearance of the nanoradiosensitizers needs to be systematically evaluated before their clinical applications. Except for the molecule mechanisms of nanoradiosensitization mentioned above, other mechanisms, such as the induction of the anti-tumor immune response, provide novel strategies for the design of nanoradiosensitizers. In addition, multifunctional nanoradiosensitizers combining with other anticancer therapies can offer new opportunities for synergistic therapy.

## Figures and Tables

**Figure 1 nanomaterials-10-00504-f001:**
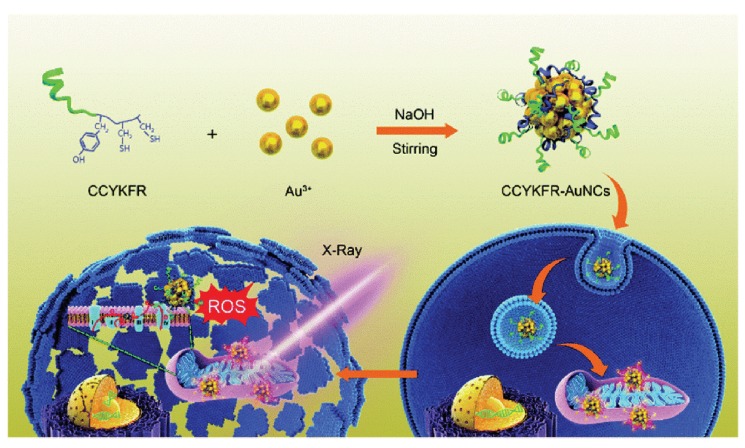
Enhanced radiosensitivity of cancer cells by gold nanoclusters (AuNCs) with mitochondria targeting. Peptide-templated AuNCs were synthesized through a green synthetic route, featured with highly efficient co-localization onto mitochondria after endocytosis. Under 4 Gy X-ray irradiation, peptide (CCYKFR)–AuNCs can introduce the burst of mitoROS and cell death. Reproduced with permission [30]. Copyright The Royal Society of Chemistry, 2017.

**Figure 2 nanomaterials-10-00504-f002:**
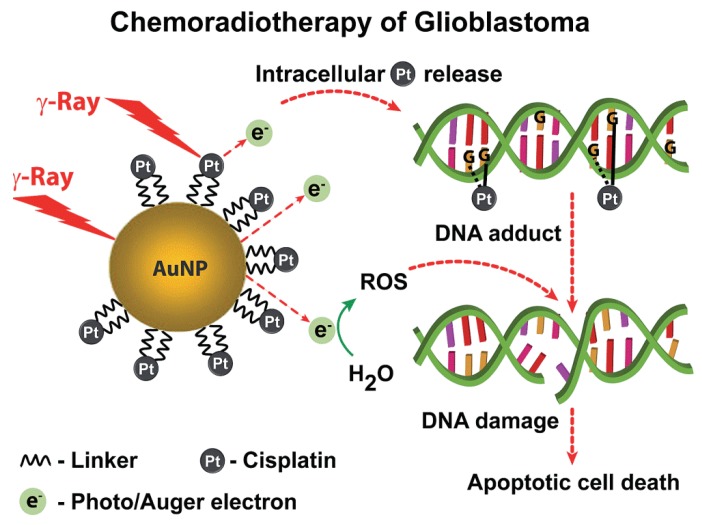
Multifunctional nanosphere for combined chemo-radiotherapy. Principle of nanosphere action based on gold and platinum mediated radiosensitization and cisplatin induced genotoxic damage. Reproduced with permission [85]. Copyright The Royal Society of Chemistry, 2014.

**Figure 3 nanomaterials-10-00504-f003:**
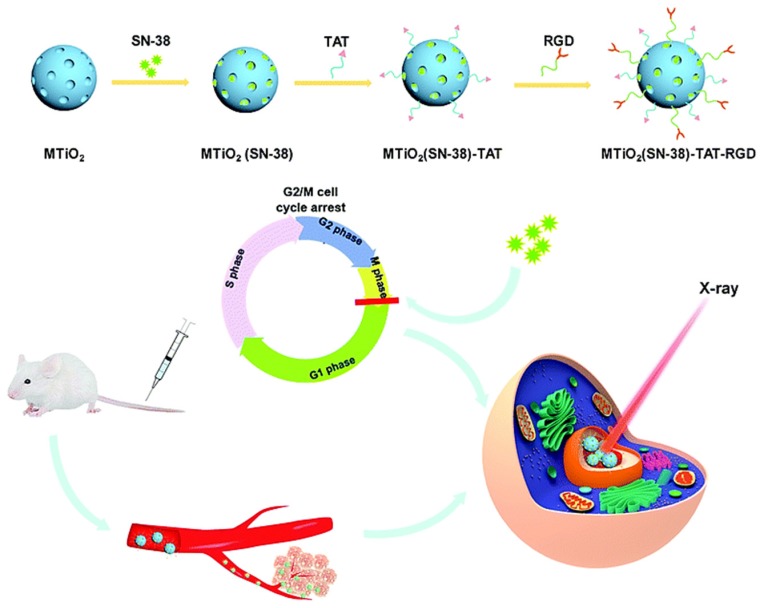
Preparation of MTiO_2_(SN-38)- transactivator of transcription (TAT)-Arg-Gly-Asp (RGD) nanoparticles (NPs) and its application for enhanced radiotherapy. Reproduced with permission [127]. Copyright The Royal Society of Chemistry, 2019.

**Figure 4 nanomaterials-10-00504-f004:**
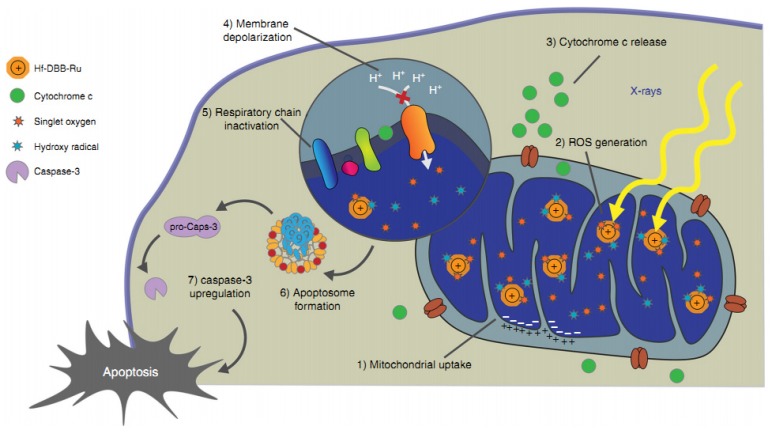
Mitochondria-targeted radiotherapy (RT)-radiodynamic therapy (RDT) mediated by Hf-DBB-Ru. Hf-DBB-Ru was internalized by tumor cells efficiently and enriched in mitochondria due to dispersed cationic charges in the nMOF framework. Hf_6_ SBUs preferentially absorb X-rays over tissues to enhance RT by sensitizing hydroxyl radical generation and enable RDT by transferring energy to Ru(bpy)_3_^2+^-based bridging ligands to generate singlet oxygen. The RT-RDT process trigger mitochondrial membrane potential depolarization, membrane integrity loss, respiratory chain inactivation, and cytochrome c release to initiate apoptosis of cancer cells. Reproduced with permission. Copyright Springer Nature, 2018.

**Figure 5 nanomaterials-10-00504-f005:**
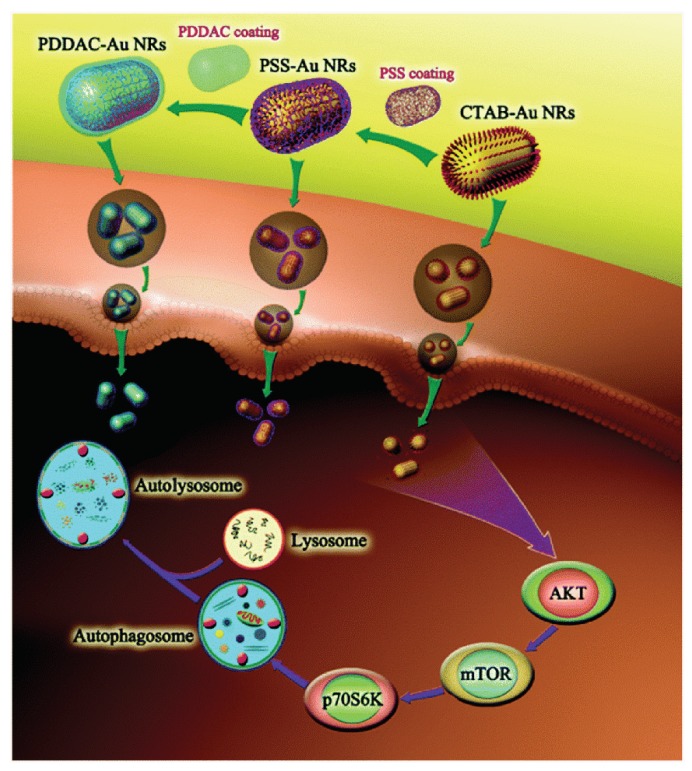
The mechanism and signaling pathways in autophagy induced by cetyl trimethyl ammonium bromide gold nanorods (CTAB-GNRs) and polystyrene sulfonate gold nanorods ( PSS-GNRs). Reproduced with permission [185]. Copyright The Royal Society of Chemistry, 2015.

**Figure 6 nanomaterials-10-00504-f006:**
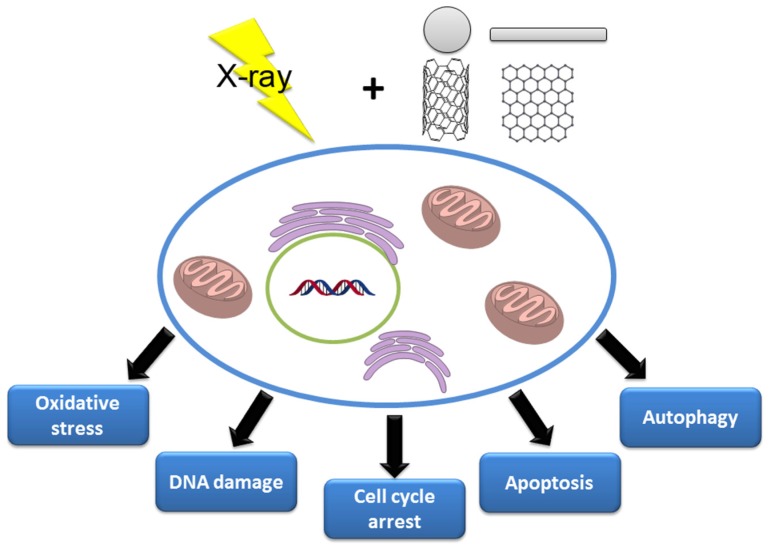
Biological mechanisms involved in NPs’ radiosensitization.

**Table 1 nanomaterials-10-00504-t001:** Summary of nanoradiosensitizers based on oxidative stress mechanism.

Composition	Size (nm)	Surface Chemistry	Cell Line/Model	Source Energy	DEF/SER/Effect	Ref.
Gold	20	Polyethylene glycol (PEG)	MDA-MB-231	2, 4, 6, 8, 10 Gy; 320 kVp	Increased cytotoxicity	[24]
Gold	10	Amino-PEG	A549	25 keV/μm protons or 225 kV X-rays	1.14 (protons), 1.22 (X-rays)	[25]
Gold	2	Levonorgestrel	EC1 cells/EC1 tumor-bearing nude mice	4 Gy	Increased cytotoxicity	[26]
Gold	3	Histidine	U14	6 Gy	1.54	[27]
Gold	14	Thio-glucose	SK-OV-3	90 kVp, 6 MV	Increased cytotoxicity	[28]
Gold	6	Octaarginine (R8)	LS180	6 MV X-rays, 0–10 Gy	1.59	[29]
Gold	3	Mitochondria-targeting peptide (CCYKFR)	MCF-7	4 Gy	1.31	[30]
Gold-TiO_2_	18	Triphenylphosphine (TPP)	MCF-7/4T1 tumor-bearing Balb/c mouse	4, 6 Gy	Tumor volume inhibition	[31]
Gold	54	HS-PEG-CH_3_, HS-PEG-NH_2_, HS-PEG-folate (FA), cell-penetrating peptide TAT	KB/U14 tumor-bearing mice	4 Gy	TAT > FA > NH_2_ > CH_3_	[32]
Gold	2	Glutathione (GSH)	MCF-7	2.25 Gy	Reduced survival rates	[33]
Gold	GNPs (53), gold nanospikes (GNSs, 54), gold nanorods (GNRs, 50 × 12)	PEG	KB	4 Gy	1.62 (GNPs), 1.37 (GNSs), 1.21 (GNRs)	[34]
Fe_3_O_4_-Gold	12.5	1-methyl-3-(dodecylphosphonic acid) imidazolium bromide	MCF-7, A549	1, 2, 3 Gy	Reduced survival rates	[35]
α-Fe_2_O_3_@Au	49	Dopamine	4T1	6 Gy	Suppressed tumor growth	[36]
Au@Se	120	Positively-charged chitosan	A375	4 Gy	Suppressed tumor growth	[37]
TiO_2_-gold	70.1	PEG	SUM159	5, 10 Gy	Suppressed tumor growth	[27]
Silver	N.A.	N.A.	HepG2	6 Gy	1.98	[38]
Silver	15	Polyvinylpyrrolidone	U251	4 Gy	Increased cell death	[39]
Silver	20–30	Polyvinylpyrrolidone	MDA-MB-231	0–4 Gy	Increased cell death	[40]
Iron oxide	10	Cetuximab	U87MGEGFRvIII	10 Gy	Increased cell death	[41]
Ferrocene	75	PEG	4T1	4 Gy	Increased cell death	[42]
Selenium	27.5	GSH	MCF-7	0–8 Gy	Increased cell death	[29]
Selenium	200	Chitosan, transferrin	C6, A375	2 Gy	Increased cell death	[43]
Selenium	<40	PEG	HeLa	8 Gy	Increased cell death	[44]
Gadolinium oxide	3	N.A.	A549, NH1299, NH1650	2 Gy	Increased cell death	[45]
Gd_2_O_3_, CeO_2_-Gd	<100	N.A.	U-87MG	3 Gy	Increased cell death	[46]
Gd-doped titania	20	4-carboxybutyl triphenylphosphonium bromide	MCF-7	4 Gy	1.7	[47]
Cerium oxide	N.A.	N.A.	L3.6pl, Panc1	5 Gy	Increased cell death	[48]
CuO	5.4	N.A.	MCF-7 cells, U14 tumor-bearing nude mice	0–8 Gy	Increased cell death	[49]
Silicon	<5	2-methyl 2-propenoic acid methyl ester	rat glioma C6 cells	0–3 Gy	Promoted ROS production	[50]
Silicon	N.A.	NH_2_	MCF-7	3 Gy	Promoted ROS production	[51]
C_60_	90–100	N.A.	B16, SMMU-7721	0–10 Gy	Increased cell death	[52]
Hydrogenated nanodiamonds	16	N.A.	Caki-1, ZR75.1S, ZR75.1R	4 Gy	Increased cell death	[53]
MWCNTs loaded with ruthenium polypridyl complex	Diameter: 225	NH_2_	R-HepG2	8 Gy	Increased cell death	[54]

Abbreviations: DEF, dose enhancement factor; SER, sensitization enhancement ratio; N.A., not available; Ref., references.

**Table 2 nanomaterials-10-00504-t002:** Summary of nanoradiosensitizers based on DNA damage mechanism.

Composition	Size (nm)	Surface Chemistry	Cell Line/Model	Source Energy	DEF/SER/Effect	Ref.
Gold	2	N.A.	MDA-MB-231	2 Gy	Enhanced DNA damage	[80]
Gold	12	PEG	U251	4 Gy (in vitro), 20 Gy (in vivo)	1.3	[81]
Gold and superparamagnetic iron oxide nanoparticles (SPION)-loaded micelles	100	Dodecanethiol, oleic acid	U251, U373	4 Gy	Enhanced DNA damage	[82]
Au@Se	120	Positively-charged chitosan	A375	4 Gy	Suppressedtumor growth	[37]
Nanogel containing GNPs	N.A.	PEG	SCCVII	15 Gy	Increased cell death	[83]
Gold	4.5	DNA	U251MG-P1	1–5 Gy	Increased cell death	[84]
Gold	50	Cisplatin	GBM cell lines (S2)	10 Gy	Inhibited cell proliferation	[85]
Gold	30	Herceptin	SK-BR-3	0–1 Gy	Enhanced DNA damage	[86]
Gold	30	Herceptin	MDA-MB-361	0–7 Gy	1.6	[87]
Gold	5	Doxorubicin	HeLa	3 Gy	Increased cell death	[88]
Gold	18	Chitosan, Doxorubicin	MCF-7	0.5, 1, 3 Gy	Increased cell death	[89]
Gold	8	Goserelin, PEG	PC3	4 Gy	Increased cell death	[90]
Gold	14	Polyethylenimine	A712	80 mGy/min	Increased cell death	[91]
Gold	<2	GSH, BSA	HeLa cells and U14 tumor-bearing nude mice	0–8 Gy	1.3 (GSH), 1.21 (BSA)	[92]
Silver	20–30	Polyvinylpyrrolidone	MDA-MB-231	0–4 Gy	Increased cell death	[40]
Silver	20	Epidermal growth factor receptor-specific antibody	nasopharyngeal carcinoma epithelial (CNE) cells	0, 2, 4, 6, 8 Gy	1.4	[93]
Iridium	<5	RGD, transactivator of transcription (TAT)	4T1	4, 6, 8 Gy	Increased cell death	[94]
Gadolinium-doped ZnO	9	N.A.	SKLC-6	0–8 Gy	1.47 (10 μg/mL), 1.61 (20 μg/mL)	[95]
Iron-oxide	10	Oleic acid	WEHI-164	2 Gy	Inhibited cell proliferation	[96]
SPION	4–6	Chitosan, PEG, PEI, siRNA	UW228, Res196	2 Gy	Increased cell death	[97]
Thulium oxide	40–45	N.A.	9L gliosarcoma	0–8 Gy	1.32	[98]
Tantalum oxide	80	Doxorubicin	4T1	0–6 Gy	Increased cell death	[99]
ZnO	7	N.A.	SKLC-6	2 Gy	1.23 (10 μg/mL), 1.31 (20 μg/mL)	[100]
Polymeric NPs containing camptothecin	20–30	N.A.	HT-29	0–6 Gy	2.2	[101]
Polymer NPs loaded with JNK inhibitor	<100	N.A.	Lewis lung carcinoma (LLC) cells	0–10 Gy	Suppressedtumor growth	[102]
PLGA NPs containing DNA double-strand repair inhibitors	87	PEG	H460	0–8 Gy	Increased cell death	[74]
PEI carrying DSB bait	140	Folate	PC-3, 22Rv1	0–8 Gy	Increased cell death	[103]
BSA NPs loaded organic selenocompound	255	Folate	HeLa	8 Gy	Increased cell death	[104]

Abbreviations: DEF, dose enhancement factor; SER, sensitization enhancement ratio; N.A., not available; Ref., references.

**Table 3 nanomaterials-10-00504-t003:** Summary of nanoradiosensitizers based on cell-cycle arrest mechanism.

Composition	Size (nm)	Surface Chemistry	Cell Line/Model	Source Energy	DEF/SER/Effect	Ref.
Gold	5	N.A.	Huh7, HepG2	5 Gy of *γ*, 5 GyE of neutron radiation	1.16–1.80	[117]
Gold	50	N.A.	HTB-72	0–4 Gy	Increased cell death	[118]
Gold	11	Glucose	DU-145	2 Gy	Growth inhibition	[119]
Gold	44 × 15	Arg-Gly-Asp peptides (RGD)	A375	0–8 Gy	1.35	[120]
Gold	55	Epidermal growth factor receptor (EGFR) antibody	HeLa	5, 10 Gy	Increased cell death	[121]
Gold	16, 49	Glucose	MDA-MB-231	0–10 Gy	1.86 (49 nm), 1.49 (16 nm)	[54]
Gold-silver	150	*l*-ascorbic acid	HepG2	0–10 Gy	1.8	[122]
Graphene	Lateral size: 18	N.A.	SW620, HCT116 c	3, 6 Gy	Increased cell death	[123]
Bi_2_O_3_	45	Hyaluronic acid	SMMC-7721	0–9 Gy	Increased cell death	[124]
Iron-oxide	10	Oleic acid	WEHI-164	2 Gy	Inhibited cell proliferation	[96]
Gadolinium oxide	2–5	N.A.	A549, NH1299, NH1650	0–4 Gy	1.10 (A549), 1.11 (NH1299), 1.20 (NH1650)	[125]
Titanate nanotubes	10	N.A.	SNB-19, U87MG	0.5, 1, 2, 5, 10 Gy	Increased cell death	[126]
TiO_2_	45	SN-38, nucleus-targeting moieties	4T1-Luc	4, 6 Gy	Inhibited cell proliferation	[127]
Hydroxycamptothecin-loaded micelles	132	Folate	HeLa	Total dose of 20 Gy	Inhibited tumor growth	[128]
Ceria	3–5	Neogambogic acid	MCF-7	0–8 Gy	Inhibited cell proliferation	[129]
Micelles	85	Paclitaxel	Lewis lung carcinoma cells	Total dose of 12 Gy	Inhibited tumor growth	[130]
PLGA	200–500	Paclitaxel	HepG2, HeLa	0–10 Gy	Increased cell death	[131]
PLGA	500	Paclitaxel	MCF-7	0–10 Gy	Increased cell death	[132]
PLGA	130–150	Docetaxel	A549, CNE-1	0–8 Gy	1.68 (A549), 1.61 (CNE-1)	[133]
Fe_3_O_4_@ZnO	<200	Doxorubicin, transferrin receptor antibody	SMMC-7721	3 Gy	Inhibited tumor growth	[134]

Abbreviations: DEF, dose enhancement factor; SER, sensitization enhancement ratio; N.A., not available; Ref., references.

**Table 4 nanomaterials-10-00504-t004:** Summary of nanoradiosensitizers based on apoptosis mechanism.

Composition	Size (nm)	Surface Chemistry	Cell Line/Model	Source Energy	DEF/SER/Effect	Ref.
Gold	5	N.A.	HSC-3	2, 4, 8 Gy	Increased cell death	[139]
Gold	18	BSA	U87	0–8 Gy	1.37	[140]
Au@Fe_2_O_3_	44	Folate	KB	2, 4 Gy	Increased cell death	[141]
Gold	55	EGFR antibody	HeLa	5, 10 Gy	Increased cell death	[121]
GNRs@mSiO_2_	76 × 33	RGD	MDA-MB-231	0–10 Gy	1.52	[142]
Gold	20	Cyclic RGD	NCI-H446 tumor-bearing mice	5 Gy	Inhibited tumor growth	[143]
Gold	8, 50, 187	BSA	H22 hepatoma-bearingmice	5 Gy	1.93 (8 nm), 2.02 (50 nm)	[144]
Gold	5, 12, 27, 49	PEG	HeLa	0–8 Gy	1.41 (5 nm), 1.65 (12 nm), 1.58 (27 nm), 1.42 (49 nm)	[145]
GNPs+17-AAG	N.A.	Folate	HCT-116	2 Gy	Increased cell death	[146]
Gold	78	Cisplatin	B16	0–8 Gy	1.29	[147]
Gold	30	Cetuximab	A431 tumor xenograft	25 Gy	Inhibited tumor growth	[148]
Gold	56	Anti-c-Met antibodies	CaSki	0–10 Gy	Increased cell death	[149]
Gold	50–70 × 35	siRNA	HNSCC	2.5 Gy	Increased cell death	[150]
Silver	15	Citrate	U251	4 Gy	1.64	[151]
Gadolinium based NPs	3	N.A.	SQ20B	0–4 Gy	Increased cell death	[152]
Selenium	80	PEG	A549	N.A.	Increased cell death	[153]
QDs	48	Amine	H460	6 Gy	Increased cell death	[154]
Cu_2_(OH)PO_4_	5	Poly(acrylic acid) sodium	HeLa	N.A.	Inhibited tumor growth	[155]
nMOFs	<100	N.A.	MC38	0–16 Gy	2.68	[156]
Mesoporous silica	<200	Valproic acid	C6, U87	0–8 Gy	1.71	[157]
Silica	40	Hyperbranched polyamidoamine	SK-BR3	0–8 Gy	Inhibited tumor growth	[158]
Black phosphorus QDs	3	PLGA	A375	0–4 Gy	Inhibited tumor growth	[159]
Black phosphorus nanosheets	300 × 25	N.A.	A375	4 Gy	Inhibited tumor growth	[160]
Dendrimer	20	N.A.	OCM-1	2 Gy	Increased cell death	[161]
PEG-Pep-PCL	85	Docetaxel	BGC823, SGC7901, MKN45, GES-1	0–8 Gy	1.24	[162]
Cationic copolymer	N.A.	Plasmid encoding *HGFK1* gene	U87, U251	0–10 Gy	1.38	[163]
HSA	180–220	Antibody, miRNA	U87MG, LN229	0–10 Gy	1.64 (U87MG), 1.25 (LN229)	[164]

Abbreviations: DEF, dose enhancement factor; SER, sensitization enhancement ratio; N.A., not available; Ref., references.

**Table 5 nanomaterials-10-00504-t005:** Summary of nanoradiosensitizers based on autophagy mechanism.

Composition	Size (nm)	Surface Chemistry	Cell Line/Model	Source Energy	DEF/SER/Effect	Ref.
Silver	27	N.A.	U251	0–8 Gy	1.78	[177]
Fe_3_O_4_@Ag	11	N.A.	U251	0–8 Gy	1.80	[178]
Gadolinium oxide	2–5	N.A.	A549, NH1299, NH1650	0–4 Gy	1.10 (A549), 1.11 (NH1299), 1.20 (NH1650)	[125]
Copper cysteamine	N.A.	N.A.	SW620	1–4 Gy	Increased cell death	[179]
CuO	5.4	N.A.	MCF-7 cells, U14 tumor-bearing nude mice	0–8 Gy	Increased cell death	[49]
FePt/GO	3	N.A.	H1975	0–8 Gy	Inhibited tumor growth	[180]

Abbreviations: DEF, dose enhancement factor; SER, sensitization enhancement ratio; N.A., not available; Ref., references.

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
