# Peer review of "The Rational Design and Biological Mechanisms of Nanoradiosensitizers"

_nanomaterials, 2020, doi:10.3390/nano10030504_

Round 1

Reviewer 1 Report

Overall this manuscript is well-written and provides an excellent summary of advancements made so far in the field of nanotechnology-based radiosensitization. Specific points that should be considered in a minor revision of this manuscript are mentioned below.

1.The authors present a good coverage of mechanisms based on oxidative stress, DNA damage, cell cycle, and cell death (apoptosis/autophagy etc.), however, the addition some discussion dealing with "anti-tumor immune response during nanotechnology-based radiosensitization" would be helpful if information is available

2 Using the glioblastoma (figure 2) example needs further support on delivery methods. How are these radiosensitizers made bioavailable and biocompatible?

  1. Section-7 (conclusions and perspectives): A paragraph on the limitations and future research directions of this technology could be added.
  2. The authors mentioned radiomodification activity by nanoceria. In this context, they should cite the paper “ Phys. Chem. C, 2017, 121, 20039 – 20050. This work illustrates nanoceria as a SOD mimetic and estimation of Michailis-Menten constants have been reported. This work demonstrates the fact that even though the same synthesis method and oxidizers are used, just the change in the anion of the precursor salt can extensively change the physicochemical properties of nanoparticles, especially its SOD-mimetic activity.

Author Response

Response to Reviewer 1 Comments

Overall this manuscript is well-written and provides an excellent summary of advancements made so far in the field of nanotechnology-based radiosensitization. Specific points that should be considered in a minor revision of this manuscript are mentioned below.

  1. The authors present a good coverage of mechanisms based on oxidative stress, DNA damage, cell cycle, and cell death (apoptosis/autophagy etc.), however, the addition some discussion dealing with "anti-tumor immune response during nanotechnology-based radiosensitization" would be helpful if information is available

We appreciate the reviewer’s comment. We have added discussions about “anti-tumor immune response during nanotechnology-based radiosensitization” in line 584-587. “Finally, the anti-tumor immune response induced during nanotechnology-based radiosensitization has shown great effect in reducing the side-effects of radiation[187] and enhancing the abscopal effects of radiationtherapy[188], which present novel strategies to design radiosensitizers.”

  1. Using the glioblastoma (figure 2) example needs further support on delivery methods. How are these radiosensitizers made bioavailable and biocompatible?

We appreciate the reviewer’s comment. In paper 88 (figure 2), the radiosensitization activity was assessed in patient-derived GBM cell lines in vitro not in vivo.

  1. Section-7 (conclusions and perspectives): A paragraph on the limitations and future research directions of this technology could be added.

We appreciate the reviewer’s comment. We have added a paragraph about the limitations and future research directions of NPs-based radiosensitization (line 588-598). “Despite the rapid advancement over the past years, more attention could be paid to the following issues. The radiosensitization effects of NPs are strongly related to their preparation procedures, particle sizes, geometries, surface chemistries and biosafety. At presence, most of the nanoradiosensitizers are developed with high-Z metal elements, which are usually undegradable and cause the biosafety concerns. Therefore, the impact of the physicochemical properties on the biocompatibility, biodistribution, biodegradability, and clearance of the nanoradiosensitizers need to be systematically evaluated before their clinical applications. Except the molecule mechanisms of nanoradiosensitization mentioned above, other mechanisms, such as the induction of the anti-tumor immune response, provide novel strategies for the design of nanoradiosensitizers. In addition, multifunctional nanoradiosensitizers combining with other anticancer therapies can offer new opportunities for synergistic therapy.”

  1. The authors mentioned radiomodification activity by nanoceria. In this context, they should cite the paper “J. Phys. Chem. C, 2017, 121, 20039 – 20050. This work illustrates nanoceria as a SOD mimetic and estimation of Michailis-Menten constants have been reported. This work demonstrates the fact that even though the same synthesis method and oxidizers are used, just the change in the anion of the precursor salt can extensively change the physicochemical properties of nanoparticles, especially its SOD-mimetic activity.

We appreciate the reviewer’s comment. We have added this paper in line 151-153. “In contrast, another work showed that cerium nanoparticles (CNPs) exhibited higher SOD-mimetic activity, which could be modulated by the change in the anion of the precursor salt [50].”

Reviewer 2 Report

This manuscript comprehensively summarized the applications of different kinds of nanoparticles to serve as radiosensitizers in cancer radiotherapy. The authors clearly integrated the biological mechanism that various NP exerted in cancer cells upon radiotherapy process and gave us a very good principle or guideline for the design of effective functional NPs in clinical radiotherapy. However, the colors used in some figures seemed to be over-saturated thus make it difficult to be distinguished between the organelles and the NPs in the presented figures.

Author Response

  1. This manuscript comprehensively summarized the applications of different kinds of nanoparticles to serve as radiosensitizers in cancer radiotherapy. The authors clearly integrated the biological mechanism that various NP exerted in cancer cells upon radiotherapy process and gave us a very good principle or guideline for the design of effective functional NPs in clinical radiotherapy. However, the colors used in some figures seemed to be over-saturated thus make it difficult to be distinguished between the organelles and the NPs in the presented figures.

We appreciate the reviewer’s comment. We have slightly reduced the saturation of Figure 1 and 5.